# Validation of the Perceived Phubbing Scale to the Argentine Context

**DOI:** 10.3390/bs13020192

**Published:** 2023-02-20

**Authors:** Edgardo Etchezahar, Marian Durao, Miguel Ángel Albalá Genol, Mariela Muller

**Affiliations:** 1Faculty of Education, International University of Valencia, 46002 Valencia, Spain; 2Faculty of Psychology, University of Buenos Aires, Buenos Aires C1207ABQ, Argentina; 3National Scientific and Technical Research Council (CONICET), Buenos Aires C1040AAH, Argentina

**Keywords:** phubbing, smartphone, social exclusion, fear of missing out, FoMO, validation

## Abstract

In recent years, the increased use of mobile devices has changed social dynamics. One such change is the rise of phubbing, described as the behavior of ignoring someone in order to pay attention to one’s cell phone. The purpose of this research was to validate the Perceived Phubbing Scale (PPS) and examine its relationship with other psychological variables. An Argentine sample was composed of 1608 participants aged between 18 and 65 (M = 45.59; SD 14.03), with 51.6% identifying as female. They were provided with a phubbing scale, along with scales to measure emotional disconnection, fear of missing out (FoMO) and social exclusion, and a sociodemographic questionnaire. The results indicated that the PPS showed an adequate fit to the data, based on a structure of one factor (X^2^ _(20)_ = 259.353, *p* < 0.001; CFI = 0.958; IFI = 0.958; RMSEA = 0.089) and the internal consistency (α = 0.93), resulting in a nine-item scale. Participants with high or medium levels of phubbing showed a tendency to suffer FoMO or feel socially excluded or socially isolated. We did not find differences in the levels of phubbing related to the participant’s age, gender, or socioeconomic level. It is possible to conclude that PPS can be used as a reliable measure to evaluate perceived phubbing in Argentina. Implications of the variables studied are discussed as possible predictors of phubbing and are to be considered in its approach.

## 1. Introduction

In recent years, communication and social functioning through the use of mobile devices has increased dramatically in day-to-day life, resulting in the practice of specific attitudes and behaviors that have an impact on social relationships [1,2]. In this regard, our postpandemic world has brought about many changes, among these working from home, new business models, and new offers for virtual college courses. With these new uses of technologies, there comes a risk of lowered performance, disrupted communication, limited collaboration, and difficulties when clarifying roles and expectations and the frequent and sometimes excessive use of various technologies is the cause of interpersonal and emotional problems [2,3,4,5].

By the end of 2021, the world’s population of smartphone users reached four billion, with their daily usage growing steadily [6]. Many people have been using smartphones to work, study, share information, and develop social relationships more frequently in the postpandemic era [7]. In this regard, excessive cell phone use is likely to cause symptoms of overdependence, tolerance, and withdrawal, in addition to other problems generally caused by behavioral addictions [8].

Specifically, according to Argentina’s Observatorio de Tendencias Sociales y Empresariales (observatory of social and business trends) (2019), 57.7% of those surveyed stated that in face-to-face conversations, in the family environment, and also among friends, they have the habit of not paying attention because they are looking at their phones [9]. These changes and implications of cell phone use during social encounters have been described as “phubbing”, which is defined as the behavior of ignoring someone (or “snubbing” them) in order to pay attention to one’s cell phone, thus interfering with interpersonal communication and eye contact [10,11]. The act of phubbing (which turns one person into a phubber and the person being ignored into a phubbee) is now a part of modern society and a phenomenon that will continue to expand [12]. A relevant study question, therefore, relates to how behaviors linked to phubbing are considered acceptable or normative in certain social environments. To this end, it is necessary to design instruments to study the extent of phubbing, in relation to other psychosocial variables, in specific contexts such as Argentina.

### 1.1. The Evaluation of Perceived Phubbing

Since the original validation of the Perceived Phubbing Scale at the international level, several online validations of the instrument focused on its structural analysis and on its relationship with other variables [13]. For example, Chotpitayasunondh and Douglas designed the Generic Scale of Being Phubbed (GSBP), comprised of 22 items related to the perception of being phubbed. It included participants aged between 18 and 63, in two different samples. Along the same lines, another scale designed earlier by Chotpitayasunondh and Douglas (measured the frequency and duration of the act of ignoring others to focus on phone use (phubbing) and the experience of feeling ignored while another is using their phone (being phubbed) [10]. Subsequently, the Parental Phubbing Scale was developed, composed of seven items, which assesses the perception of phubbing [14]. The scale was validated in the Italian context with an adolescent population who were asked about their perception of phubbing between their parents. It is worth mentioning the work of Karadag et al., who developed a scale to assess phubbing with 10 items measuring two factors: 1. Communication Disorders (α = 0.87), which assesses whether participants often disrupt their existing communications to deal with their cell phones in a face-to-face communication setting, and 2. Telephone Obsession (α = 0.85), which assesses whether participants need their cell phone in environments that lack face-to-face communication [15]. The participants were university students. A noteworthy aspect of this assessment is that the second factor (as its name suggests), rather than assessing phubbing, is aimed at investigating smartphone addiction.

Lastly, David and Roberts’s Perceived Phubbing Scale, originally designed on the basis of previous work, is one of the most widely used today [11,13]. The structure in the original validation, and in successive studies, employed one factor, replicating adequate validity, reliability, and consistency in the descriptive statistics in relation to the nine items that measured perceived phubbing exerted by others. Specifically, the confirmatory factor analysis led to evidence of adequate construct validity (X2 (27) = 80.02, *p* < 0.001; CFI = 0.97; NFI = 0.96; RMSEA = 0.08), with adequate reliability (α = 0.93). In this study, two groups were established among the participating sample (145 adults) based on the level (high or low) of phubbing they perceived to have suffered. Higher levels of conflict in interpersonal relationships involving cell phone use were evident in the “low phubbing” group, compared with the “high phubbing” group, particularly with respect to individuals’ feelings of social exclusion, the fear of missing out (FoMO), and social disconnection; these are discussed below [16]. This scale was widely documented in previous studies in at least 20 countries, including Brazil, China, Croatia, Ecuador, India, Italy, Netherlands, Pakistan, Poland, Portugal, Slovenia, Spain, Turkey, Ukraine, and the USA, among others [17].

### 1.2. Relationships between Perceived Phubbing, Feelings of Social Exclusion, FoMO, and Social Disconnection

Phubbing and its perception in various everyday environments play a fundamental role in the feeling of social exclusion and in the use of digital devices and social media [11]. The present paper uses as a reference David and Roberts’s sequential model, which hypothesizes that when a person feels phubbed, he or she will have greater feelings of social exclusion and will also show a greater need for attention and a more intense use of social media. This model was developed from the Social Exchange Theory and the Kardefelt–Winther model of compensatory internet use (2014) [17]. The authors state that in order to improve their sense of inclusion, people who suffer phubbing paradoxically seek to socialize more intensely within the digital realm rather than interact face-to-face. The possibilities of connection facilitated by smartphones are hampered by the impact of phubbing on our well-being [18]. The perception of being ignored or phubbed is associated with feelings of exclusion and one’s own need for attention, which, in turn, are associated with higher levels of social media use and psychological discomfort [19,20]. The feeling of exclusion affects our ability to regulate our own emotions, thoughts, and behaviors while compromising our capacity for reasoning and adequate perception of time [21,22]. This way, people can turn to their smartphones and social media to connect with others and alleviate their pain. The subjective pain caused by social exclusion can be comparable to physical pain, since it activates the same areas of the brain [23]. Conversely, social connectedness (i.e., the perception of having close relationships with society and a sense of belonging) has effects that mitigate physical pain [24]. For example, David and Roberts hypothesized and found that this need to feel included is associated with being phubbed [16]. The main concern is to regain a sense of inclusion, and, to this end, one may seek to compensate through excessive use of social media [17]. Although the original studies did not show significant differences in relation to participants’ gender, other studies did show that those who engage in phubbing do so to a greater extent to women, and for this reason women develop greater feelings of social exclusion than do men [12,25].

While phubbing is a relatively new concept, there is evidence of a significant association with the fear of missing out (FoMO) [10,11,26]. Additionally, FoMO has been found to motivate individuals to seek out socially inclusive experiences [27]. It has also been shown that FoMO may be a predictor of phubbing and excessive use of social media among adolescents [28]. Other studies suggest that people experiencing FoMO will strive to feel more socially included by continually checking their smartphones in order to alleviate FoMO-related anxiety [29]. There are also studies that link phubbing perceived by children with social disconnection from one or both parents, which leads to a greater socioemotional distancing from the family environment [14]. Social disconnection, competition, and loneliness lead to negative emotions that affect physical and mental health [30]. In this context, phubbing constitutes a situation in which people are forced to negotiate between immediate, face-to-face contact and the interruption produced by other contact via digital media [31].

The main objective of this paper was to analyze the psychometric properties of the Perceived Phubbing Scale in a sample of Argentine participants and study its relationship with feelings of social exclusion, FoMO, and social disconnection [11]. In the present investigation, two hypotheses were proposed. On the one hand, the process of adaptation and validation of the scale will form an instrument with adequate psychometric characteristics in the Argentine context. On the other hand, based on the perceived phubbing levels, differences will be found in FoMO, feelings of social exclusion, and social disconnection.

## 2. Materials and Methods

### 2.1. Participants

A geolocalized online questionnaire was administered, with stratified sampling based on the geographical regions of Argentina. The complete and valid protocols totaled 1608 cases (with a sampling error of ±2.5% and a confidence level of 95%). Of the participants, 51.6% (*n* = 829) identified as female, 48.3% (*n* = 776) identified as male, and 0.2% (*n* = 3) identified as nonbinary. The average age of the participants was 45.59 years (SD = 14.03), with an age range between 18 and 70 years. With respect to educational level, 3.8% of the sample had primary education (complete and incomplete), 29.6% had secondary education (complete and incomplete), 33.6% had tertiary education (complete and incomplete), and 33% had complete university studies.

### 2.2. Measures

Self-report measures were employed using a battery of assessment instruments consisting of:

Phubbing Scale. We used the scale developed by David and Roberts, which is composed of nine items that assess how often people use their smartphones while spending time with others (i.e., friends, neighbors, family, etc.) (e.g., “People who I spend time with often glance at their cell phone when talking to me,” “When their cell phone rings or beeps, they pull it out even if we are in the middle of a conversation,” “When I spend time with people, they keep their cell phone where they can see it”). The response format ranges from 1 = Never to 5 = All the time. The internal consistency was adequate (α = 0.910) [11].

FoMO scale. To assess the construct, we proceeded to complete the validation of the original version of the scale Przybylski, composed of 10 items that determine dimension 1, FoM NI (e.g., “I fear my friends have more rewarding experiences than I do”) (α = 0.813) and dimension 2, FoM SO (e.g., “It bothers me when I miss an opportunity to meet up with friends”) (α = 0.780) [32]. Each item was rated on a Likert scale with five anchors, ranging from 1 = Strongly disagree to 5 = Strongly agree (the same response format was used for the other scales used in this study). The higher the score in both dimensions, the higher the levels of FoMO.

Feelings of Social Exclusion Scale. The scale originally developed by Cheung and Choi (2000) and reformulated by David and Roberts is composed of six items that inquire about feelings of social exclusion (e.g., “To what extent when spending time with other people do you experience feelings of being ignored?” “To what extent when spending time with other people do you experience feelings of being excluded?” To what extent when spending time with other people do you experience feelings of being rejected?”) [16,33]. The response format is five anchors, ranging from 1 = Not at all to 5 = Very much. The internal consistency was adequate (α = 0.827).

Social Connectedness. We used the Lee and Robbins (1994) Social Connectedness scale composed of eight items that cover different aspects of belongingness: connectedness, affiliation, and companionship (e.g., “Even around people I know, I don’t feel that I really belong,” “I catch myself losing all sense of connectedness with society,” “I don’t feel I participate with anyone or any group”) [34]. The scale portrays a general emotional distance between self and others that may be experienced even among friends or close peers. Each item was rated on a Likert scale with five anchors, ranging from 1 = Strongly disagree to 5 = Strongly agree (the same response format was used for the other scales used in this study). The internal consistency was adequate (α = 0.766).

Sociodemographic data questionnaire: Information on gender, age, self-perceived socioeconomic level, and highest level of education was collected from the participants.

### 2.3. Procedure

People who met the criteria of age (over 18 years of age) and geographic region were invited to participate via social media, based on the quotas stipulated for the sample distribution. Participants were previously informed, at the start, about the purpose of the study and the institution responsible for it and were provided with a contact e-mail address in case they required further information. Additionally, they were informed that the data collected in this study would be used only for academic–scientific purposes and would be protected in accordance with Argentine National Law 25,326 on the protection of personal data.

### 2.4. Data Analysis

The statistical analyses that guided the development of this study were conducted using SPSS for Windows software version 19.0 and EQS 6.1 for the development of the Confirmatory Factor Analysis (CFA) of the Phubbing scale structure [35,36]. In all cases, the normality of the scales was tested for the use of parametric statistics. Furthermore, descriptive statistics were analyzed for each of the items that make up the definitive version of the scale (mean, standard deviation, skewness, and kurtosis). Next, the internal consistency of the instrument was analyzed through the use of Cronbach’s alpha. Lastly, with the aim of assessing criterion validity, three levels of the Perceived Phubbing Scale were divided, analyzing the differences with FoMO, feelings of exclusion, and social disconnection through the use of the ANOVA one-way and eta^2^ for the effect size. All data were fitted to a normal distribution to report the means and standard deviations.

## 3. Results

First, the descriptive statistics and reliability of the Phubbing Scale items were analyzed in the Argentine context (Table 1).

As shown in Table 1, the levels of skewness and kurtosis were adequate for all items. Regarding the reliability, the total alpha was 0.879, the elimination of item 7 improved the alpha to 0.910 and the item-total correlation for this item was very low (>0.35) (Hair et al., 2016), so we discarded this element. The percentage of variance accounted for on the eight-item scale was 61.47%.

In order to test the internal validity of the Phubbing Scale, a confirmatory factor analysis was performed, and the evidence confirmed an adequate construct validity (X2 (20) = 259.353, *p* < 0.001; NNFI = 0.941; CFI = 0.958; IFI = 0.958; RMSEA = 0.089)—except for the RMSEA, which barely exceeds the cutoff criteria—and an adequate reliability (α = 0.93). All indicators are adequate for the unidimensional model for the data collected in this study of phubbing.

Next, after analyzing the descriptive statistics of the items, and the reliability and validity of the Phubbing Scale, we proceeded to analyze the possible relationships with age, gender, educational level, and social class of the study participants. No statistically significant differences were observed between phubbing and any sociodemographic variables.

Perceived Phubbing Levels and Differences in FoMO, Feelings of Exclusion, and Social Disconnection

The participants were then divided into three groups (low, medium, and high) based on their scores on the Phubbing Scale (percentile 33 and 66), following the recommendations of David and Roberts (2020). Subsequently, we proceeded to analyze whether there were statistically significant differences between the levels of phubbing and the FoMO 1 dimension. (F = 16.184; *p* <.001; partial eta^2^ = 0.023) (Figure 1) and FoMO 2 (F = 8.624; *p* < 0.001; partial eta^2^ = 0.013) (Figure 2).

As seen in Figure 1, participants who reported having high-to-medium levels of phubbing in their social interactions showed significantly higher levels of FoMO 1, compared with those exposed to low levels.

Similarly, as shown in Figure 2, there are significant differences with respect to FoMO 2 between those who have low levels of phubbing and those who have medium and high levels.

The same process was then conducted with feelings of social exclusion (*F* = 19.110; *p* < 0.001; partial eta^2^ = 0.030) (Figure 3) and with levels of social disconnection (*F* = 14.468; *p* < 0.001; partial eta^2^ = 0.023) (Figure 4).

As seen in Figure 3, regarding feelings of social exclusion (Figure 3), significant differences were observed in the three groups, corresponding to the low, medium, and high levels. In contrast with the results for FoMO, in the results for feelings of social exclusion, the differences between the three evaluated levels of perceived phubbing are relevant for this analysis (and not only the comparison between high–medium levels with respect to low levels).

As for the levels of social disconnection (Figure 4), two groups were formed: a first group with participants with low and medium levels of phubbing and a second group with participants with high levels of phubbing. Inversely to the differences found in FoMO (Figure 1 and Figure 2), in social disconnection the results show a comparative analysis approach (with significant differences) between low–medium levels and high levels of phubbing perceived by the participants.

## 4. Discussion

The main objective of this paper was the validation of the Perceived Phubbing Scale developed by David and Roberts in the Argentine context [11]. First, the descriptive indicators were adequate for the nine items of the scale, with adequate item-total correlation and reliability indices, and with no improvement as a result of the elimination of items [37]. Also, consistent with the original validation, the CFA results showed an adequate fit for the one-factor model, identifying a dimension for the perceived phubbing construct (the RMSEA barely exceeds the cutoff criteria, a result that should be taken into account in future studies). This validation differs from those related to other instruments. On the one hand, it is a short instrument that assesses the level to which the participants are phubbed in their various social interactions. However, many of the instruments designed and validated focus on evaluating the phubbing practiced (and not perceived) by individuals in their social contexts. This approach to analysis and the results obtained coincide with those found in other validations of the scale in different contexts [10,11,13,38]. Therefore, the present study offers an instrument of perceived phubbing, validated to the Argentine context. It is important to point out that longer scales such as the Generic Scale of Being Phubbed (GSBP) by Chotpitayasunondh and Douglas, made up of 22 items, were evidenced as a limitation and a factor to be improved through the design of shorter instruments adapted to the current survey modality [17,38]. In this sense, a limitation of the present study was not comparing both scales to corroborate the greater effectiveness of a short instrument such as the adapted one.

Regarding the relationships between perceived phubbing and sociodemographic variables such as age, gender, and self-perceived social class, as also reported in the original study, no statistically significant differences were found in this study. However, on other occasions research did find that women were more frequently involved in and suffered from phubbing than were men, which may happen because men are more familiar in some contexts with face-to-face meetings than are women [1,11,12,25]. A possible limitation to be improved in future studies is related to the need to provide the study of the variables analyzed with a crosscultural character.

In relation to the contrast of the external validity of the instrument, we analyzed the differences in other variables related to perceived phubbing, based on its levels (high, medium, and low). In the case of FoMO-NI, the significantly lower levels shown among those who felt less phubbed indicate less fear and anxiety about missing out on new experiences and information when living in less technology-centric environments during social interactions (e.g., not ignoring or disregarding as often during social interactions: meetings, meals, etc.). Likewise, in the case of FoMO-SO, similar results are produced, with those who perceive themselves to be less phubbed in their environments also showing less anxiety and fear of missing out on social opportunities and experiences, manifesting a lower impulse to use technologies. These findings are consistent with other studies that have associated FoMO closely with the habit of phubbing and with the experience of being phubbed [10,26,28]. In this way, FoMO could be an indicator of the need for attention that David and Roberts place as an immediate consequence to perceived phubbing and the feelings of social exclusion it provokes, as well as the increased intensity of social media use that stems from this [16,28,39]. An innovation of the study was to evaluate FoMO with a valid instrument made up of two dimensions, since each of these must continue to be contrasted with other variables derived from the digital field.

From this perspective, with respect to feelings of social exclusion, significant differences were also evident based on the three levels of perceived phubbing analyzed. The results showed that those who reported being less phubbed also felt less socially excluded. Respectively, those who reported medium levels of phubbing felt significantly more socially excluded, as did those who suffered high levels, results consistent with David and Roberts’s model and its application also to couples [16]. The behaviors that make a person feel phubbed, therefore, have an impact on such a significant aspect as the feeling of social exclusion that stems from it, which may have psychological and emotional consequences, and also lead to social disconnection [18,20,30]. A limitation to improve in future studies is related to evaluating more indicators that corroborate the levels of social exclusion and social disconnection felt in the participants.

In this sense, the levels of social disconnection were also significantly different as a function of the perceived level of phubbing, which shows that this variable is related not only to feelings of social exclusion but also to effective social disconnection. Therefore, the distance and social disconnection experienced by those who exhibited high levels of phubbing in their environments were significantly higher than in those who experienced medium and low levels, respectively. This represents an empirical innovation, because although the David and Roberts model has been contrasted in some contexts, in countries like Argentina hardly any research had been carried out that explored these variables based on perceived phubbing. In this regard, among those who suffer it more frequently, phubbing could be promoting a greater social distancing toward the people around them, including family and friends [14,40]. Moreover, this is related to their feeling of belonging and connection with society in general and, therefore, with their psychosocial well-being and the coexistence they develop [12].

After the recent period of pandemic and confinement experienced globally due to COVID-19, some of the practices and behaviors derived from our social interaction and the use of technologies have increased. All this has led to a notable deterioration in the psychological and social well-being of the general population. Argentina being no exception, which poses the challenge of designing and adapting assessment instruments that guarantee the rigorous study of the impact of phubbing [40]. Given the novel nature of this first validation of the Perceived Phubbing Scale, there are certain limitations in the study that could be overcome with various future considerations. First, we suggest delving deeper into the relationship between FoMO and perceived phubbing, because due to the novelty of the concept of phubbing, there are a limited number of studies that analyze its scope [29]. Second, we suggest incorporating other psychosocial variables in relation to perceived phubbing (linked to the three levels of analysis posited by David and Roberts, 2017: feelings of social exclusion, need for psychological care, and abusive use of social media), as well as other levels of analysis. Likewise, we recommend exploring in a differential way the levels of perceived phubbing in seldom-studied samples such as older adults, given that technologies and their impact on social interactions are also present in the elderly [41].

If we want to live together in more cohesive and inclusive societies, we need to continue developing instruments such as the Perceived Phubbing Scale that can help us understand the impact of technology use on people’s psychological and social well-being and in their daily lives.

## Figures and Tables

**Figure 1 behavsci-13-00192-f001:**
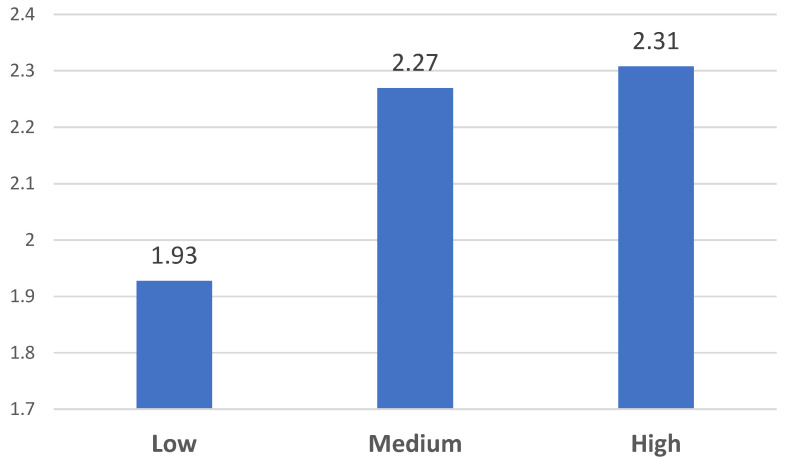
Levels of FoMO 1 differences based on perceived phubbing levels.

**Figure 2 behavsci-13-00192-f002:**
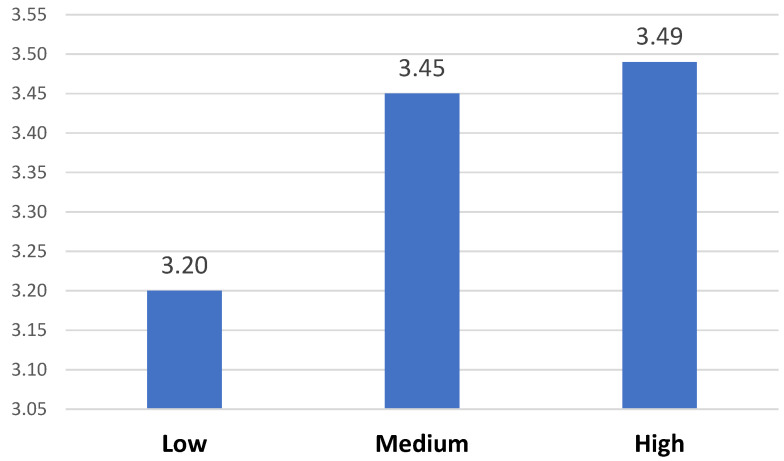
Levels of FoMO 2 based on perceived phubbing levels.

**Figure 3 behavsci-13-00192-f003:**
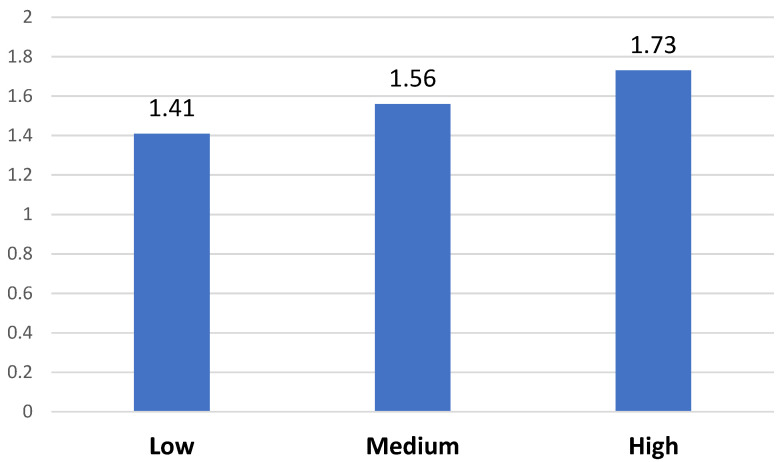
Feelings of social exclusion differences based on perceived phubbing levels.

**Figure 4 behavsci-13-00192-f004:**
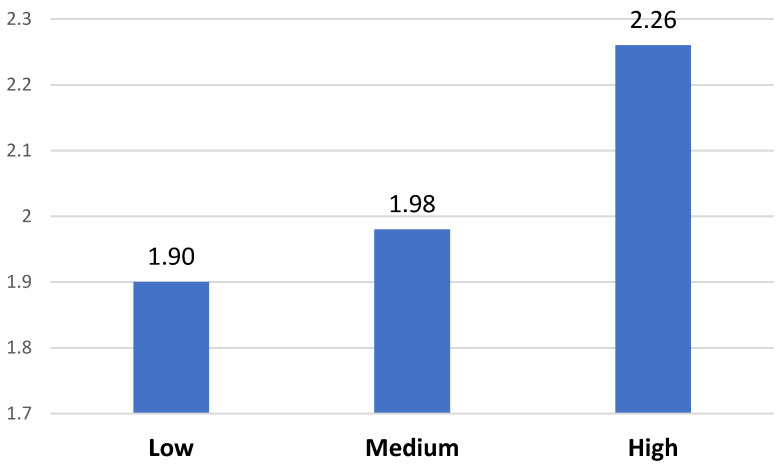
Social disconnection differences based on perceived phubbing levels.

**Table 1 behavsci-13-00192-t001:** Descriptive Statistics and Reliability of Phubbing Items.

	M	SD	S	K	r.it	a-i
1. When I’m spending time with people over a meal, they pull out and check their cell phones.	3.17	1.032	0.148	−0.501	0.686	0.853
2. When a cell phone rings or vibrates, people look at it even when we are in the middle of a conversation.	3.35	1.016	0.019	−0.482	0.646	0.857
3. When I spend my free time with other people, they use their cell phones.	3.36	1.023	0.043	−0.618	0.742	0.848
4. People I spend time with often look at their cell phones when we are talking.	3.09	1.031	0.183	−0.531	0.785	0.844
5. When I spend time with people, they keep their cell phones where they can see them.	3.65	1.026	−0.169	−0.755	0.629	0.859
6. People use their cell phones when we are talking in person.	2.94	1.037	0.191	−0.599	0.739	0.848
7. People never have their cell phones in their hands when they are with me.	2.30	0.957	1.183	1.296	−0.040	0.910
8. When I go out with other people, they use their cell phones at some point during our time together.	3.21	1.001	0.287	−0.563	0.653	0.856
9. If there is a pause in a conversation, people check their cell phones.	3.34	1.043	0.030	−0.569	0.706	0.851

Note. M = Mean; SD = Standard Deviation; S = Skewness; K = Kurtosis; r.it = item-total correlation; a-i = Cronbach’s Alpha if the item is deleted.

## Data Availability

Data are available upon request.

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
