# Peer review of "Validation of the Perceived Phubbing Scale to the Argentine Context"

_behavsci, 2023, doi:10.3390/bs13020192_

Round 1

Reviewer 1 Report

ID: behavsci-2080731

Title: Adaptation and Validation of the Perceived Phubbing Scale to the Argentine Context

Thank you for providing a chance to review this manuscript.

Comment: Rejection or Major Revision.

Detailed information:

Title: I think there is no adaptation process in this study.

Abstract

The abstract should contain the background, purpose, method, result and conclusion, especially the method and result should be written in detail. Please rewrite the abstract.

Introduction

Line 96-108, Page 2-3: Please provide more evidence that the PPS scale has achieved cross-cultural adaptation and validation in other cultures.

Line 155-158, Page 4: What is the hypothesis of this study?

Overall: The author spends a lot of space describing the research background and the development of the PPS scale. Please simplify.

Materials and Methods

Phubbing Scale

Line 163-169, Page 4:

1) “Lastly, David and Roberts's (2020) Perceived Phubbing Scale, originally designed on the basis of previous work (Roberts & David, 2016) is one of the most widely used today.” How many versions of the PPS scale are there? Why did the author use the 2017 version?

2) The title of this article is "Adaptation and Validation of the Perceived Phubbing Scale to the Argentine Context." Has the author adapted PPS scale? If so, please explain the adaptation process in detail.

3) How are these scales investigated? Online or offline?

Procedure

Line 199-206, Page 4-5:

1) Please specify the inclusion and exclusion criteria.

2) The specific information of participants should be written detailly in the method, including the number of cases, regions, etc. And are the participants representative?

Data Analysis

Do the data conform to normality?

Overall: With all due respect, the method was poorly written, and many important information was not shown, such as the survey method and participants' information. Please read the published articles, and rewrite the “Materials and Methods”!

Results

Subjectively perceived anxiety

Line 227-229, Page 5: “the elimination of item 7 improved the alpha to .910, and the item-total correlation for this item was very low (> .35) (Hair et al., 2016) so we discarded this element.” How does the reliability change after item 7 deletion?

Line 239-240, Page 6: RMSEA is greater than the acceptable value.

Line 241-245, Page 6: What are the general characteristics of the participants? Please present it as a table. Moreover, what are the scale recovery rate and effective rates? These are important results!

Line 249-250, Page 6: “The participants were then divided into three groups (low, medium, and high) based on their scores on the Phubbing scale.” What were the groups based on?

Figure 1-4: These results only explain the differences between the three groups, but cannot prove the “Relationships between Perceived Phubbing, Feelings of Social Exclusion, FoMO, and Social Disconnection”, and further correlation analysis and multiple regression analysis are needed.

Discussion

Line 278-280, Page 8: “The main objective of this paper was the adaptation and validation of the Perceived Phubbing Scale developed by David and Roberts (2020) in the Argentine context.” 2017 version or the 2020 version? Refer to line 163.

Line 289-292, Page 8: “It is important to point out that longer scales such as the Generic Scale of Being Phubbed (GSBP) by Chotpitayasunondh and Douglas (2018), made up of 22 items, were evidenced as a limitation and a factor to be improved through the design of shorter instruments adapted to the current survey modality.” This study did not carry out such verification.

Please state the innovations and limitations of this study.

With all due respect, I did not even finish my reading. I am feeling that there's a lot of information missing here. First, reading more articles from the TOP health quality journals, to learn the formats, expressions, and of great importance—logic, might help a lot before revising this “MANUSCRIPT”. Then, focus on the sections of “Materials and Methods” and “Results”, which should be described in detail. Furthermore, there are many basic errors in this article, such as grammar, punctuation and so on. Last but not least, finding a native English speaker to improve the writing can considerably improve the quality.

Thank you and my best,

Your reviewer

Reviewer 2 Report

This is an interesting article. It would be good if the authors pay attention to the following minor points

- Some of the studies (related to the previously developed scales) in the introduction didn't include sample age.

- The manuscript needs some editing (especially quotation marks)

- It might be useful to put some research questions at the end of the introduction.

- Why was the age of 18 set to be the criterion?

- For Table 1, it’s better to write the what the statistical symbols in the tables mean as the reader can’t interpret what these symbols mean.

- No need for Table 2, this can be described in the text.

- What was the criterion used to determine the three levels of participants (3.1)?

- Why didn’t the authors make use of the other scales to conduct a criterion-related validity?

- The authors didn’t make use of the sample size with which several analyses could have been conducted in relation to the scale invariance in terms of gender, age levels, and other related demographic variables.

- The discussion is good but it might be supported to talking about how the validation was different than other instruments, specifically the one they adapted in details.

Reviewer 3 Report

Dear authors,

In the first place, I would like to congratulate you. I think the topic you have choosen is very interesting and absolutly actual. However, I have found some aspect that I think you should try to modify in orther to improve the quality of the manuscript. 

In the first place I have not found explicitly what the gap is in the literature, and what is the foundation of this research. Please add this to the intro.

- Introducction: In line number 31 you use a reference from 2017 to talk about the pandemic situation, please, pick up a reference post pandemic. As well as that, in line number 52, you use a reference from 2015 to talk about moder societies, please, try to find a more current one. As for part 1.1. The Evaluation of Perceived Phubbing, I find interesting and well explained the first parragrahp, however, from line 72 to the end of this part, there is too much information from other studies such us the properties of other scales, etc, that it does not fit well in here, it is not apropiated to talk about this methodological aspects in the introducction. Try to simplify this whole part. In line 118 there is a point between inclusion and people, review it please. 

- Method: the description of the materials are correct, however I suggest you to include the alfa de cronbach of each scale. 

- Results: the psychometric qualities of the instrument are correctly calculated. However, from point 3.1 the wording of the results is very poor. They should describe in greater depth and clarity whether or not relationships were found between Phubbing and the different variables. The figures that have been used are not intuitive enough and without a text that complements and explains them, they remain empty. I recommend that you extend the wording of this part and, if necessary, delete the figures or improve them.

- Dicussion: the discussion is correct. I suggest you pay more attention to the conclusion. Try to conclude with more force and extolling the main findings of your research and why it is relevant.

- References: a review of all references is required. Remember that when using apa 7, you should not put the word doi, but the link directly.

Round 2

Reviewer 1 Report

ID: behavsci-2080731

Title: Validation of the Perceived Phubbing Scale to the Argentine Context

The authors have modified most of the comments, and I accept replies from the authors in most cases. In order to further improve the quality of the manuscript, there are still some small problems to be improved.

Comment: Minor Revision.

Detailed information:

Abstract

Specific values of statistical results should be supplemented.

Materials and Methods

Data Analysis

The results are expressed as mean and standard deviation, so do all the data conform to normal distribution?

Results

1) Why did the authors delete Table 2?

Figure 1-4: These results only explain the differences between the three groups, but cannot prove the “Relationships between Perceived Phubbing, Feelings of Social Exclusion, FoMO, and Social Disconnection”, and further correlation analysis and multiple regression analysis are needed. I still recommend further analysis.

Thank you and my best,

Your reviewer

Reviewer 3 Report

Dear authors,

I would like to congratulate you. It is perceptible that they have applied all the changes that I suggested to them. I believe that after this substantial improvement in content, the quality of the manuscript is adequate to be published in this journal. Receive cordial greetings.
